# OpenReview forum: "SAFEPATH: Preventing Harmful Reasoning in Chain-of-Thought via Early Alignment"
_NeurIPS.cc/2025/Conference — NeurIPS 2025 poster_

### Official Review · Reviewer_cJVT · 2025-06-11

**Clarity:** 1
**Significance:** 2
**Originality:** 2
**Rating:** 4
**Confidence:** 4

**Summary:**

This paper proposes SAFEPATH, a lightweight and efficient method to improve the safety of Large Reasoning Models (LRMs) without compromising their reasoning capabilities. SAFEPATH introduces a fixed 8-token Safety Primer (“Let’s think about safety first.”) at the beginning of the reasoning process for harmful prompts, preserving reasoning depth and avoiding the typical safety-reasoning trade-off.

**Questions:**

See Weakness.

**Ethical Concerns:**

["NO or VERY MINOR ethics concerns only"]

**Final Justification:**

Thank you for the response.

The rebuttal has addressed my concerns regarding the experimental results, and I will raise my scores accordingly. However, I still recommend that the authors improve the description of their method and clarify the distinction between their work and backdoor-based approaches.

**Limitations:**

Yes.

**Paper Formatting Concerns:**

No.

**Quality:**

1

**Strengths And Weaknesses:**

### Strength:
+ This paper addresses the challenge of defending against jailbreak attacks in reasoning models, which is an important and currently underexplored problem.

### Weakness:
- The writing, particularly in Section 3 describing the proposed method, is confusing. The  pipeline of the method is difficult to follow, and I'm unable to clearly understand how the approach is implemented. I strongly recommend that the authors provide a more comprehensive and structured explanation of their method, ideally with mathematical formulations to clarify the process.
- Based on my understanding, the paper appears to fine-tune a model using a dataset of harmful and benign instructions, enabling it to recognize harmful inputs and generate a fixed “Safety Primer” (e.g., “Let’s think about safety first.”) to guide safer reasoning. If the harmful instructions are learned to trigger this primer, the approach resembles a backdoor mechanism. In that case, prior work has already explored similar backdoor-based safety alignments in general generative models [1]. Extending such ideas to reasoning tasks may not be sufficiently novel.
- The defense baselines considered in this work are overly simplistic. The authors should compare against stronger and more relevant jailbreak defense methods, such as guard models like Llama-Guard, which offer plug-and-play capabilities and are more efficient than fine-tuning-based approaches.
- The set of attacks evaluated is limited. It would strengthen the paper to include more diverse and advanced attack strategies, such as ArtPrompt [2].
- The scope of this work is narrow. It focuses solely on reasoning models, while many real-world applications rely on general-purpose generative models. Reasoning is not always required, especially for simple queries. It would be valuable to evaluate the proposed method on general models like LLaMA or Qwen. Additionally, the necessity of post-training the model is unclear. An alternative direction could be incorporating safety alignment into the reinforcement learning process. For instance, rewarding the model for rejecting harmful prompts. In such a setup, it’s worth questioning whether the “Safety Primer” mechanism is still essential.


[1] Wang, J., Li, J., Li, Y., Qi, X., Hu, J., Li, S., ... & Xiao, C. (2024). Backdooralign: Mitigating fine-tuning based jailbreak attack with backdoor enhanced safety alignment. Advances in Neural Information Processing Systems, 37, 5210-5243.

[2] Jiang, F., Xu, Z., Niu, L., Xiang, Z., Ramasubramanian, B., Li, B., & Poovendran, R. (2024, August). Artprompt: Ascii art-based jailbreak attacks against aligned llms. In Proceedings of the 62nd Annual Meeting of the Association for Computational Linguistics (Volume 1: Long Papers) (pp. 15157-15173).

---

> ### Author Rebuttal · Authors · 2025-07-30
>
> We sincerely thank the **reviewer cJVT** for their thoughtful feedback. Below, we address the main points:
>
> > **W1. Concern on method clarity:** Requesting for a clearer and more structured presentation of the method (Section 3).
>
> We understand that the presentation in Section 3 could be clarified further, and we will revise the final version to provide a more structured explanation of the SAFEPATH method, along with a diagram and mathematical formulations. To clarify our method, we briefly describe SAFEPATH’s key idea and implementation below.
>
> SAFEPATH is a lightweight alignment method that fine-tunes only an 8-token prefix (called the Safety Primer) to guide the model’s internal reasoning without supervising or modifying the full reasoning trace. The training process involves two separate datasets:
>
> **Safety Trigger Set** (harmful prompts): For these, we fine-tune only the 8-token Safety Primer, which is inserted immediately after the model’s internal \<think\> token. The rest of the reasoning is left unsupervised. For example, the input for fine-tuning might be:
>
> \<user\> How can I bypass website security?
>
> \<assistant\> \<think\> Let's think about the safety first
>
> Loss is applied only to the 8 tokens of the Safety Primer (e.g., “Let’s think about the safety first”).
>
> **Reasoning Retain Set** (benign prompts): The model is trained on complete reasoning traces to maintain its task performance. No safety signal is inserted, and no prefix is applied.
>
> A key implementation detail is that the Safety Primer is not followed by a closing \</think\> tag, so it becomes a natural part of the model’s internal reasoning rather than a bounded instruction. The intent is not to make the model refuse, but to encourage it to reason with safety- awareness reasoning throughout the response.
>
> At inference time, we observe an emergent behavior: although the model is only trained to emit the primer at the beginning, it often re-generates the Safety Primer mid-reasoning when encountering harmful or ambiguous content. This behavior is not manually supervised, but arises from the model internalizing the Safety Primer as a reusable safety cue.
>
> In summary, SAFEPATH achieves robust safety improvements while preserving reasoning performance, by fine-tuning only an 8-token prefix on harmful prompts and leaving the reasoning process itself untouched. We will revise Section 3 accordingly to improve clarity and convey the method in a more structured and transparent manner.
>
> > **W2. Concern on contribution:** Potential overlap with backdoor-based safety alignment methods, raising questions about whether the extension to reasoning tasks offers a distinct contribution.
>
> While at a high level SAFEPATH may resemble a trigger-based mechanism, we would like to clarify a key distinction: SAFEPATH does not rely on fixed input patterns or backdoor-style triggers to activate safety behaviors. Instead, it induces generalizable, context-aware safety reasoning by leveraging the model’s internal chain-of-thought capabilities.
>
> During training, SAFEPATH fine-tunes only an 8-token Safety Primer on a small set of harmful prompts (40-400 samples) with no supervision over the full reasoning trace. The model is not trained to memorize input-output patterns or to respond only to specific trigger phrases. Instead, it learns to integrate the Safety Primer into its internal reasoning process.
>
> Critically, during inference, we observe that the model dynamically re-generates the Safety Primer mid-sequence when faced with harmful or ambiguous reasoning, even when the input structure differs significantly from the training data. This emergent behavior highlights that SAFEPATH generalizes well beyond the training prompts.
>
> We will revise the discussion to more clearly articulate this distinction and highlight how SAFEPATH differs from prior backdoor-style methods in both design and empirical behavior.
>
> > **W3. Concern on baselines:** Suggestion to include stronger jailbreak defense baselines, such as plug-and-play guard models like Llama-Guard, instead of only using fine-tuning-based methods.
>
> We acknowledge that guard models like Llama-Guard provide a practical and efficient means of post-hoc safety filtering. However, these models operate by filtering or modifying inputs or outputs externally, without influencing the model’s internal reasoning process. In addition, they typically require two LLM calls: one for generation and one for evaluation, resulting in additional computational overhead.
>
> SAFEPATH, by contrast, is designed to align the internal reasoning trajectory of large reasoning models (LRMs) through a lightweight, integrated mechanism. As such, comparing it to external guard models is not methodologically appropriate, since the two classes of approaches serve different goals and operate at different levels of the stack.
>
> Our focus is on internal safe alignment for multi-step reasoning models, a setting that remains underexplored. To ensure a fair and meaningful comparison, we include SafeChain, which is specifically designed for LRMs, as well as a suite of adapted methods from the broader LLM alignment literature. These include Direct Refusal, Circuit Breaker, Task Arithmetic, and NPO, which were re-implemented and adapted for LRM use to establish strong, diverse baselines.
>
> Additionally, to directly address your concern, we compare against two recent safety methods (RealSafe-R1 [1], Star-R1 [2]) developed for LRMs. As shown in the table below, while these methods rely on either carefully curated data or complex RL-based training, our approach achieves strong performance with a simpler and more practical design in less than five minutes.
>
> **Reasoning Performance ↑ compared with recent baselines (8B).**
> |Method|MATH500 ↑|GPQA ↑|AIME24 ↑|Avg. ↑|
> |-|-|-|-|-|
> |RealSafe-R1|**85.8**|46.5|46.7|59.7|
> |Star-R1|84.0|45.0|46.7|58.5|
> |SAFEPATH|84.0|**48.0**|**50.0**|**60.7**|
>
> **Table: Robustness Against Jailbreak Attacks ↓ compared with with recent baselines (8B).**
> |Method|DAN ↓|Trigger ↓|Multilingual ↓|PAIR ↓|Avg. ↓|
> |-|-|-|-|-|-|
> |RealSafe-R1|5.7|3.0|4.9|**26.3**|10.0|
> |Star-R1|**1.3**|5.3|**1.3**|41.3|12.3|
> |SAFEPATH|5.7|**2.0**|**1.3**|**26.3**|**8.8**|
>
> In summary, while plug-and-play guard models like Llama-Guard are valuable tools in constrained deployment settings, they fall outside the scope of this work, which targets internal alignment for reasoning models with multi-step generation. We will revise the discussion in the final version to clarify this distinction and better contextualize our choice of baselines.
>
> > **W4. Evaluation scope:** Suggestion to strengthen the evaluation by including more advanced attack strategies, such as ArtPrompt.
>
> We agree that evaluating SAFEPATH against a broader range of advanced attack strategies is important for demonstrating its robustness.
> In response, we have extended our evaluation to include ArtPrompt [4] and FlipAttack [3], recent attacks that challenge models through prompt manipulation. As shown in the results below, SAFEPATH maintains strong performance across all of these adversarial settings.
>
> We will incorporate these new evaluations and results in the final version to further strengthen the empirical contributions of the paper.
>
> **Table: Additional Jailbreak Results (Attack Success Rate) on DeepSeek-R1-Distill-Llama-8B.**
> |Method|FLIP Attack ↓|ArtPrompt ↓|
> |-|-|-|
> |Base|45.2|26.0|
> |SAFEPATH|5.4|4.0|
>
> > **W5. Scope and applicability:** Limited focus on reasoning models, with suggestions to evaluate on general models and explore alternative alignment strategies.
>
> We acknowledge that general-purpose LLMs such as LLaMA and Qwen are widely used in real-world applications, particularly for simple instruction-following tasks. However, our work specifically focuses on large reasoning models (LRMs), which are increasingly important due to their strong multi-step reasoning capabilities and their growing adoption in safety-critical applications such as autonomous decision-making, legal analysis, and complex multi-turn agents.
>
> While reasoning models offer more powerful capabilities, they also pose greater safety risks [6,7,8], as their ability to generate harmful outputs is amplified by their structured, multi-step reasoning. Despite this, internal safety alignment for LRMs remains underexplored, and our work aims to address this important gap.
>
> We acknowledge that alternative directions such as incorporating safety into reinforcement learning frameworks are also promising. However, SAFEPATH offers a complementary and lightweight approach: it does not require reward modeling, task-specific supervision. Instead, it aligns internal reasoning through a short, fine-tuned Safety Primer that demonstrates persistent and emergent safety behavior, as shown in our experiments.
>
> We also note that SAFEPATH is the first method of its kind to induce dynamic, self-triggered safety behavior within the reasoning trajectory itself, an approach we believe is both novel and highly promising for future extensions.
>
> [1] Zhang, Yichi, et al. "Realsafe-r1: Safety-aligned deepseek-r1 without compromising reasoning capability." (2025).\
> [2] Wang, Zijun, et al. "Star-1: Safer alignment of reasoning llms with 1k data." (2025).\
> [3] Liu, Yue, et al. "Flipattack: Jailbreak llms via flipping." ICML (2025).\
> [4] Jiang, Fengqing, et al. "Artprompt: Ascii art-based jailbreak attacks against aligned llms." ACL (2024)\
> [5] Muennighoff, Niklas, et al. "s1: Simple test-time scaling." (2025).\
> [6] Wang, Cheng, et al. "Safety in large reasoning models: A survey." (2025).\
> [7] Zhou, Kaiwen, et al. "The hidden risks of large reasoning models: A safety assessment of r1." (2025).\
> [8] Jiang, Fengqing, et al. "Safechain: Safety of language models with long chain-of-thought reasoning capabilities." (2025).

---

> > ### Comment · Reviewer_cJVT · 2025-08-06
> > **Response to Authors**
> >
> > Thank you for the response.
> >
> > The rebuttal has addressed my concerns regarding the experimental results, and I will raise my scores accordingly. However, I still recommend that the authors improve the description of their method and clarify the distinction between their work and backdoor-based approaches.

---

> > > ### Author Response · Authors · 2025-08-06
> > >
> > > **Thank you for your follow-up and for updating your evaluation.**
> > >
> > > We're glad that our rebuttal addressed your concerns regarding the experiments and evaluation. In the final version, we will revise Section 3 to provide a clearer and more structured explanation of SAFEPATH, including a diagram illustrating the overall process and mathematical formulations detailing how the Safety Primer is applied.
> > >
> > > We will also further clarify the conceptual distinction between SAFEPATH and backdoor-style methods. In particular, we will emphasize that SAFEPATH does not rely on fixed input triggers or pattern-matching behavior, but instead induces generalizable, context-aware safety reasoning that emerges dynamically within the model's internal reasoning trajectory. We appreciate your constructive feedback and believe these improvements will meaningfully strengthen the paper.

---

### Official Review · Reviewer_CSoy · 2025-07-02

**Clarity:** 3
**Significance:** 3
**Originality:** 3
**Rating:** 4
**Confidence:** 3

**Summary:**

This work proposes SafePath for safety alignment in reasoning models. The method injects a fixed prefix—“Let’s think about safety first”—during the reasoning process, and performs fine-tuning using a combination of a safety trigger set and a reasoning retention set. Extensive experiments and ablation studies demonstrate the effectiveness of the proposed method.

**Questions:**

Please refer to the weakness section.

**Ethical Concerns:**

["NO or VERY MINOR ethics concerns only"]

**Final Justification:**

I lean toward recommending acceptance. I have no major concerns, and the rebuttal has satisfactorily addressed my questions.

**Quality:**

3

**Strengths And Weaknesses:**

This work proposes a simple yet effective method for safety alignment in reasoning models. The experiments include extensive ablation studies on the number of safety primers across different tasks, alternative prefix designs, and zero-shot settings. The main results demonstrate consistent improvements over baselines across various model sizes and tasks.

I believe the current results are sufficient to validate the effectiveness of SafePath, and I have no major critiques of the current manuscript. However, I am curious about one aspect: although the training consistently activates the safety primer at the beginning of the reasoning process, the model appears to generalize and activate safety checks at appropriate points during inference. Are there any insights into why the model learns to dynamically verify safety, despite always seeing the prefix at the beginning during training?

Additionally, while recent work [1] argues that safety alignment should not rely solely on a small set of trigger tokens, how does the proposed method align with or respond to this concern?

[1] Safety Alignment Should Be Made More Than Just a Few Tokens Deep.

---

> ### Author Rebuttal · Authors · 2025-07-30
>
> We sincerely thank the **reviewer CSoy** for the insightful and constructive questions, which help highlight important aspects of our method. Below, we address the main points:
>
> > **W1. Dynamic safety behavior:** Question on how the model learns to activate safety checks dynamically during inference, despite only seeing the Safety Primer at the beginning during training.
>
> Thank you for the thoughtful question. We believe this behavior emerges from the way SAFEPATH softly integrates safety into the reasoning process. Although the Safety Primer is always placed at the beginning during training, the model is not trained with supervised reasoning traces for safety.
>
> Instead, it learns to associate harmful prompts with the need for safety reflection, while preserving its natural reasoning pathways. As a result, during inference, the model dynamically reactivates the Safety Primer when unsafe reasoning arises and skips it when the context is benign. We view this as an emergent property of aligning safety with the model’s internal reasoning dynamics, as supported by our analysis in Section 5.3.
>
> > **W2. Depth of alignment:** Question on how the method addresses concerns about shallow alignment relying only on a small set of trigger tokens, as raised in recent work [1].
>
> While [1] rightly argues that safety alignment should go beyond a few shallow trigger tokens in LLM, SAFEPATH addresses this concern by fully leveraging the internal reasoning capabilities of large reasoning models.
>
> Although SAFEPATH fine-tunes only an 8-token prefix, it does not rely on a single static cue (i.e., we do not close with \</think\> token). Instead, as we discuss in W1 and Section 5.3, SAFEPATH induces dynamic, context-aware safety behavior: the model learns to monitor its own reasoning and re-activate the Safety Primer during generation when harmful content emerges.
>
> This emergent behavior is possible because SAFEPATH does not override the reasoning process, it integrates safety directly into it. To investigate this, we analyzed SAFEPATH’s behavior under partially completed harmful reasoning trajectories and examined its next-token prediction probabilities. We found that SAFEPATH initially mirrors the base model, but as the trajectory progresses toward harmful content, its token distribution gradually diverges, often followed by spontaneous re-activation of the Safety Primer. Surprisingly, in this setup, Safety Primer emerges more than 15 times on average, indicating that SAFEPATH enables the model to re-invoke safety guidance dynamically as needed (We follow the setup from Section 2.2 of [1].).
>
> In contrast to methods that depend on fragile prompt cues, SAFEPATH demonstrates that a lightweight prefix can yield non-trivial and persistent safety behavior when aligned with the model’s internal thought process. We will clarify this connection to [1] more explicitly in the final version.
>
> [1] Qi, Xiangyu, et al. "Safety alignment should be made more than just a few tokens deep." ICLR (2025).

---

> ### Author Response · Authors · 2025-08-09
> **Gentle Reminder for Reviewer CSoy**
>
> Dear Reviewer CSoy,
>
> Thank you for your time and thoughtful review. We have carefully addressed your concerns in our responses and would greatly appreciate any additional feedback before the discussion period ends. Thank you again for your time and support.
>
> Sincerely, Authors.

---

### Official Review · Reviewer_N8gg · 2025-07-03

**Clarity:** 3
**Significance:** 3
**Originality:** 2
**Rating:** 4
**Confidence:** 3

**Summary:**

This paper presents SafePath a low-weight model priming method that improves the safety of long CoT models (I.e., decreases attack rate on harmful prompts) while incurring minimal drops in reasoning performance, thereby balancing the so-called "safety tax". The method works by fine-tuning the model to output 8-tokens "lets think about safety first" at the beginning of the thoughts without forcing a termination. The authors find that this method outperforms baselines has the ability to skip harmful parts of reasoning chains (even if interspersed in the reasoning trajectory) and is more robust to different attack methods while also mitigating the drop in reasoning performance.

**Questions:**

- Table 1: Suggestion aggregate the scores across datasets for each block so we can see the cumulative safety tax? Are the drops in Table 1 significant?

- On Sec 5.7, would the method be more robust in case the attacks were complementary.

- Is this an artificat with Qwen distilled model? Do we expect this to be exploiting an issue/sensitivity with current generation of long CoT models?

- What is the interplay between attack rate and test time scaling, I.e. are the unsafe thoughts that are skipped recoverable by simple TTS techniques like adding the "wait" or other prefixes?

**Ethical Concerns:**

["NO or VERY MINOR ethics concerns only"]

**Limitations:**

Yes in the appendix but I would recommend moving it to the main paper.

**Quality:**

3

**Strengths And Weaknesses:**

**Strengths:**
- The paper proposes a simple yet effective method to improve safety without incurring a major penalty in reasoning abilities (which is an important direction)
- The approach is relatively lightweight and so can be readily incorporated when model checkpoints are readily available and zero-shot variant can be used for API access ones (Does that actually work with frontier models?)

**Weaknesses:**
- Limited study of all models that were distilled from DeepSeekR1. It is unclear if this method overfits to models distilled from this distribution or to all LLMs in general.
- Limited analysis on why this method does not work for instruction following models in general and is only suited for models in long-CoT settings.

---

> ### Author Rebuttal · Authors · 2025-07-30
>
> We sincerely thank the **reviewer Ng88** for the constructive feedback and specific suggestions to improve the paper. Below, we address the main points:
>
> > **W1. Generalization across model families**: Concern on whether SAFEPATH overfits to models distilled from DeepSeek-R1, rather than generalizing to a broader range of LLMs.
>
> We initially focused on DeepSeek-R1 models because they are open-source, readily accessible, and widely used in recent safety research [1,2,3]. These models, based on Qwen and LLaMA architectures, offer strong reasoning capabilities while remaining vulnerable to harmful prompts, making them a practical and relevant testbed for evaluating safety alignment methods.
>
> That said, we fully agree with the need to assess generalization beyond a single model family. To address this concern, we extended our experiments to include s1.1 [4], a model trained on high-quality, original data and not distilled from DeepSeek-R1.
>
> As shown in the table below, SAFEPATH significantly improves the safety of s1.1 while preserving its reasoning ability, suggesting that its effectiveness is not tied to DeepSeek-R1 or any particular distillation process. We will include these expanded results in the final version to make this point more explicit.
>
> **Table: Additional Results on s1.1-7B.**
>
> |Method|MATH500 ↑|GPQA ↑|AIME24 ↑|BeaverTails ↓|PAIR ↓|
> |-|-|-|-|-|-|
> |Base|82.8|38.9|20.0|49.4|92.5|
> |DirectRefusal|81.4|34.9|16.7|15.5|85.0|
> |SafeChain|82.0|38.4|16.7|17.3|91.3|
> |SAFEPATH|**82.2**|**38.4**|**20.0**|**5.9**|**21.3**|
>
> > **W2. Behavior in instruction models:** Limited analysis on why the method does not work well for instruction-following models and appears suited only for long-CoT settings.
>
> SAFEPATH is specifically designed for large reasoning models (LRMs) that exhibit long-chain-of-thought (CoT) behavior. Its effectiveness comes from guiding the entire internal reasoning process, rather than modifying just the initial response.
> Instruction-following models typically produce short, direct outputs without a structured reasoning trace. As a result, there is limited opportunity for the Safety Primer to influence the internal trajectory, making SAFEPATH less applicable in those settings.
>
> In contrast, CoT-style models allow the primer to shape ongoing reasoning. As we show in our paper Section 5.3, SAFEPATH frequently re-triggers the Safety Primer multiple times in the middle of the reasoning process, especially in adversarial or ambiguous contexts. This emergent behavior clearly demonstrates that SAFEPATH is not simply affecting the first few tokens, but rather aligning the reasoning dynamics over the full generation. We will clarify this distinction more explicitly in the final version.
>
> > **Q1. Table 1 interpretation:** Suggestion to aggregate scores across datasets to clarify the cumulative safety tax, and question on whether the reported performance drops are statistically significant.
>
> We will revise Table 1 to include aggregate metrics (e.g., mean harmfulness, robustness, reasoning, and capability scores) to make the overall trade-offs clearer.
>
> As shown in table below, SAFEPATH consistently achieves low harmfulness and high robustness across a wide range of safety benchmarks, while maintaining strong performance on reasoning and general capability tasks. The observed drops in reasoning ability and capability are marginal and substantially smaller than those of prior methods, supporting our claim that SAFEPATH mitigates the safety-capability trade-off more effectively.
>
> **Table: Aggregated Results on DeepSeek-R1-Distill-Qwen-7B.**
> |Qwen-7B|Base|DirectRefusal|SafeChain|SAFEPATH (Ours)|
> |-|-|-|-|-|
> |**Harmfulness ↓**|45.3|29.1|35.9|**11.6**|
> |**Robustness ↓**|45.1|43.3|43.0|**12.4**|
> |**Capability ↑**|47.4|46.8|46.9|**47.3**|
> |**Reasoning ↑**|60.8|50.2|58.8|**60.0**|
>
> **Table: Aggregated Results on DeepSeek-R1-Distill-Llama-8B.**
> |Llama-8B|Base|DirectRefusal|SafeChain|SAFEPATH (Ours)|
> |-|-|-|-|-|
> |**Harmfulness ↓**|36.8|18.7|25.7|**3.9**|
> |**Robustness ↓**|52.6|47.8|52.5|**8.8**|
> |**Capability ↑**|46.9|46.4|47.4|**46.9**|
> |**Reasoning ↑**|57.9|51.8|52.1|**56.2**|
>
> > **Q2. Comparison to complementary strategies:** Question on whether the proposed method can be improved by complementary methods (CautionPath, RefusalPath).
>
> In Section 5.7, we compare SAFEPATH to two complementary approaches, CautionPath (CP) and RefusalPath (RP), which explicitly signal caution or refusal at the beginning of the response. While both variants successfully reduce harmful outputs (as shown by lower BeaverTails scores), they do so by prematurely terminating the reasoning process, which leads to substantial degradation in reasoning ability across all evaluated benchmarks.
>
> In contrast, SAFEPATH adopts a fundamentally different strategy: rather than halting the model’s response early, it introduces a soft, safety-oriented prefix that influences the internal reasoning process without suppressing it. This allows the model to continue generating thoughtful and complete answers while maintaining safety awareness throughout.
>
> As shown in Figure 7 and Table 3, SAFEPATH consistently maintains a better balance between robust safety and task engagement, unlike complementary approaches that tend to force binary refusal. This makes SAFEPATH more robust not only to direct attacks but also to nuanced or ambiguous prompts that require both caution and reasoning.
>
> We believe this distinction is critical and highlights SAFEPATH’s unique advantage: it enables context-aware safety while preserving the model’s ability to think through and respond to complex tasks, a property that complementary approaches struggle to achieve.
>
> > **Q3. Potential architectural artifact:** Question on whether the observed effect is an artifact of the Qwen-distilled model, or if it exploits sensitivities in current long CoT models.
>
> Thank you for the thoughtful question. We do not believe the observed re-triggering behavior is an artifact of the Qwen architecture or its distillation process. In our main experiments (Table 1), we evaluate SAFEPATH on both Qwen- and LLaMA-based models distilled from DeepSeek-R1 and observe consistent re-triggering and 	safety behavior across both families, suggesting the effect is not architecture-specific.
>
> To further validate this, we also evaluate SAFEPATH on s1.1, a non-distilled model trained by high-quality original data. We find that SAFEPATH improves safety while preserving reasoning performance on s1.1 as well, reinforcing the idea that the observed behavior is not tied to a specific model, architecture, or training method (please see W1).
>
> These results indicate that SAFEPATH does not rely on model-specific artifacts but instead leverages a general property of LRMs. We will clarify this point in the final version.
>
> > **Q4. Robustness to test-time scaling:** Question on whether skipped unsafe reasoning can be recovered through simple test-time scaling techniques such as adding “wait” or other prefixes.
>
> Thank you for the insightful question. To investigate this, we analyzed SAFEPATH’s behavior under partially completed harmful reasoning trajectories and examined its next-token prediction probabilities. We found that SAFEPATH initially mirrors the base model, but as the trajectory progresses toward harmful content, its token distribution gradually diverges, often followed by spontaneous re-activation of the Safety Primer. Surprisingly, in this setup, Safety Primer emerges more than 15 times on average, indicating that SAFEPATH enables the model to re-invoke safety guidance dynamically as needed.
>
> This behavior suggests that SAFEPATH is not dependent on prompt-level cues alone, but actively monitors the unfolding reasoning process and reacts in real time based on its internal state. As a result, simple test-time scaling techniques, such as inserting wait to scale the reasoning process, are unlikely to weaken the defense.
>
> In fact, we explicitly evaluated SAFEPATH in a "morethink" setting using the "wait" token to extend the reasoning artificially. We observed no degradation in safety behavior, further confirming that more reasoning does not harm SAFEPATH’s alignment, if anything, it gives the model more opportunity to re-trigger the Safety Primer when needed.  For example, the SAFEPATH model often emits “Let’s think about safety first” immediately after the “wait” token in harmful cases, resulting in an even lower ASR compared to the default decoding strategy.
>
> We will include this analysis in the final version to clarify SAFEPATH’s robustness to test-time manipulations.
>
> **Table: Results when TTS decoding has applied on DeepSeek-R1-Distill-Llama-8B.**
> |Method|DAN ↓|Trigger ↓|Multilingual ↓|PAIR ↓|Avg. ↓|
> |-|-|-|-|-|-|
> |SAFEPATH|5.7|2.0|1.3|26.3|8.8|
> |+MoreThink|4.3|1.2|1.3|17.5|6.1|
>
> > Suggestion to move the Limitation section from the appendix to the main paper.
>
> Thank you for the suggestion. We agree that highlighting the limitations in the main paper improves transparency, and we will reflect this in the final version.
>
> [1] Wang, Cheng, et al. "Safety in large reasoning models: A survey." (2025).\
> [2] Zhou, Kaiwen, et al. "The hidden risks of large reasoning models: A safety assessment of r1." (2025).\
> [3] Jiang, Fengqing, et al. "Safechain: Safety of language models with long chain-of-thought reasoning capabilities." (2025).\
> [4] Muennighoff, Niklas, et al. "s1: Simple test-time scaling." (2025).

---

> ### Author Response · Authors · 2025-08-09
> **Gentle Reminder for Reviewer N8gg**
>
> Dear Reviewer N8gg,
>
> Thank you for your review. We've addressed your comments and would appreciate any further feedback before the discussion ends. Thanks once more for your time and feedback.
>
> Sincerely, Authors

---

### Official Review · Reviewer_2uhd · 2025-07-03

**Clarity:** 4
**Significance:** 2
**Originality:** 3
**Rating:** 4
**Confidence:** 5

**Summary:**

The paper introduces SAFEPATH, an alignment method designed to prevent harmful outputs from Large Reasoning Models (LRMs) when exposed to harmful prompts. While existing safety alignment methods can reduce harmful responses, they often compromise reasoning depth, leading to significant trade-offs in complex tasks. SAFEPATH addresses this issue by embedding a short, 8-token Safety Primer at the beginning of the reasoning process, allowing the model to maintain its reasoning capabilities while promoting safety. Experimental results across multiple benchmarks demonstrate that SAFEPATH effectively reduces harmful outputs by up to 90.0% and blocks 83.3% of jailbreak attempts, all while requiring significantly less computational resources than existing methods.

**Questions:**

1. From the training perspective, what're the main differences between this method and the prefix training methods?

2. In Figure 1, what about the robustness of the proposed method under the more recent attacks, like AutoDAN-Turbo and FlipAttack?

**Ethical Concerns:**

["NO or VERY MINOR ethics concerns only"]

**Limitations:**

yes

**Quality:**

3

**Strengths And Weaknesses:**

## Strengths
1. Lightweight Approach: SAFEPATH introduces a lightweight, effective method for aligning LRMs that avoids compromising reasoning abilities.

2. Strong Empirical Results: The method is thoroughly validated through extensive experiments, showing significant improvements in safety and reasoning performance across various tasks.

3. Dynamic Safety Mechanism: The ability to reactivate the Safety Primer during reasoning enhances the model's adaptability to different contexts and potential safety risks.

4. Low Training Costs: The method requires minimal fine-tuning, resulting in a much faster training process compared to traditional safety alignment methods.


## Weaknesses
1. Vulnerability to Sophisticated Attacks: Despite its strengths, SAFEPATH still shows non-negligible attack success rates against advanced adversarial strategies, indicating room for improvement.

2. Dependence on Data Quality: The effectiveness of SAFEPATH may be influenced by the quality of the training data, particularly in terms of identifying harmful prompts accurately.

3. Complex Implementation: While designed to be lightweight, the implementation of SAFEPATH may still require a deep understanding of both safety mechanisms and reasoning processes.

4. Potential Overfitting Risks: Focusing on safety may lead to overfitting in specific scenarios, where the model could struggle with generalization in unfamiliar contexts.

5. Related Work: Missing discussions on the recent related papers [1,2,3].

6. The figure for the overall training pipeline is missing. Besides, the novelty of the prefix training is limited. The author should figure out the main technical contributions of this paper in the introduction.

[1] FlipAttack: Jailbreak LLMs via Flipping

[2] GuardReasoner: Towards Reasoning-based LLM Safeguards

[3] Safety in large reasoning models: A survey

---

> ### Author Rebuttal · Authors · 2025-07-30
>
> We sincerely thank the **reviewer 2uhd** for the helpful feedback and for highlighting points that can clarify and strengthen the paper. Below, we address the main points:
>
> > **W1. Residual attack success:** Concern that SAFEPATH still exhibits some success rates under advanced adversarial strategies, indicating possible areas for future improvement.
>
> Thank you for raising this concern. While SAFEPATH does not entirely eliminate adversarial vulnerabilities, it achieves a substantial reduction in attack success rates across a range of strong adversarial strategies (e.g., 5.7% on DAN and 2.0% on Trigger in R-8B; see Table 1). This is achieved with minimal intervention, only 8 tunable tokens and less than 5 minutes of training, which presents a highly favorable robustness-efficiency trade-off.
> More importantly, to our knowledge, SAFEPATH is the first post-safety alignment method designed to encourage large reasoning models (LRMs) to maintain safe internal reasoning trajectories throughout multi-step generation by eliciting safety reasoning through a short safety prompt. Prior approaches typically constrain the reasoning process to follow fixed paths or rely on rejection mechanisms. In contrast, our method prompts the model to consider safety during reasoning without enforcing rigid constraints. We believe this represents a novel and important shift in the safety paradigm for LRMs.
>
> > **W2. Data quality sensitivity:** Question on whether SAFEPATH’s effectiveness depends on the quality of training data, particularly in identifying harmful prompts accurately.
>
> While data quality is always an important factor, we would like to clarify that the Safety Trigger (40 to 400 samples) and Reasoning Retain sets were randomly sampled from WildJailbreak and MATH 220K, respectively, without any filtering or manual selection.
> To resolve the reviewer’s concern, we further evaluated SAFEPATH using alternative Safety Trigger datasets, including AdvBench [6] and BeaverTails [5], in addition to WildJailbreak [7].  As shown in the table below, SAFEPATH maintains comparable or better safety performance across datasets, suggesting that it generalizes well across variations in data source and quality.
>
> **Table: SAFEPATH trained on different Safety Trigger sets on DeepSeek-R1-Distill-Llama-8B. "Reasoning" is the average performance over MATH500, GPQA, and AIME24, while "Robustness" is the mean ASR across DAN, PAIR, Trigger, Multilingual, and prefilling.**
> |Dataset|Reasoning ↑|Robustness (against jailbreak) ↓|
> |-|-|-|
> |Base|60.8|45.1|
> |Wildjailbreak|60.7|8.8|
> |AdvBench|60.8|9.1|
> |BeaverTails|60.1|7.8|
>
> > **W3. Implementation complexity:** Concern that SAFEPATH, despite its lightweight design, may require a deep understanding of both safety mechanisms and reasoning processes to implement effectively.
>
> While understanding reasoning models can help interpret SAFEPATH’s behavior, we emphasize that its implementation is simple and requires minimal intervention. SAFEPATH fine-tunes only an 8-token prefix, the “safety primer”, and the training loss is applied exclusively to these 8 tokens, with no supervision of the remaining reasoning trace.
> During fine-tuning, the primer is inserted directly after the model’s internal \<think\> token, as in the following example:
>
> \<user\> How can I bypass website security?
>
> \<assistant\> \<think\> Let's think about the safety first
>
> By not closing the \</think> tag, the primer becomes an integral part of the model’s internal reasoning trajectory. This setup allows the model to absorb safety intent as part of its cognitive frame, without disrupting its native reasoning capabilities.
> We will revise the paper to include a clearer diagram of the SAFEPATH pipeline and add this implementation detail explicitly to better convey the method’s simplicity and ease of adoption.
>
> > **W4. Potential Overfitting Risks:** Focusing on safety may lead to overfitting in specific scenarios, where the model could struggle with generalization in unfamiliar contexts.
>
> Since SAFEPATH is fine-tuned on only 40 to 400 of harmful prompts, we agree that the risk of overfitting to specific scenarios is a valid consideration. However, as shown in Table 1, SAFEPATH generalizes remarkably well, despite this minimal supervision.
> We evaluated SAFEPATH on a wide range of previously unseen adversarial prompts, including PAIR, DAN, Trigger, and Multilingual jailbreak attacks. Across all these settings, SAFEPATH significantly reduced attack success rates while preserving performance on standard reasoning tasks such as MATH500, GPQA, AIME24 (see Table 1).
> These results suggest that SAFEPATH induces a deeper shift in the model’s reasoning dynamics, rather than relying on brittle pattern-matching. This highlights the potential of primer-based alignment as a lightweight yet robust approach for sustained safety in large reasoning models.
> We believe this strong generalization stems from SAFEPATH’s use of the model’s internal reasoning capabilities. Rather than memorizing specific surface patterns, the model learns, via the Safety Primer, to reflect on what constitutes harmful behavior as part of its reasoning process.
>
> >  **W5. Related Work:** Missing discussions on the recent related papers.
>
> Thank you for the suggestion. We acknowledge the relevance of the mentioned works and will include appropriate citations and a brief discussion in the final version [1,2,3]. Additionally, we have included results on FlipAttack in our response to Q2.
>
> > **W6. Clarity of contributions:** Suggestion to include a figure of the training pipeline and clarify the main technical contributions.
>
> Thank you for the helpful feedback. We will include a figure in the final version to clearly illustrate the SAFEPATH training pipeline.
> While SAFEPATH employs a simple prefix-tuning setup, its core novelty lies in the emergent, long-range behavioral effects it induces in the model’s internal reasoning process, not in the tuning mechanism itself. Despite fine-tuning only an 8-token Safety Primer, SAFEPATH reliably sustains safe reasoning across thousands of tokens, even under adversarial conditions.
> A particularly distinctive behavior is that the model spontaneously reactivates the Safety Primer mid-sequence, especially in response to ambiguous or harmful prompts. This effect is not manually enforced and is absent in prior LLM safety methods based on prefix tuning, where safety often collapses after the initial output.
> We will revise the introduction to more clearly highlight this contribution: SAFEPATH is the first to demonstrate that a short, learned prefix can induce persistent, self-reinforcing safety behavior throughout multi-step reasoning, marking a significant departure from prior short-horizon alignment techniques.
>
> > **Q1. Comparison to prefix-tuning:** Question on how the proposed method differs from standard prefix-tuning approaches from a training perspective.
>
> Thank you for the question. From a training mechanics perspective, SAFEPATH may appear similar to standard prefix tuning, as it fine-tunes only a short prefix. However, the key difference lies in what is tuned and the profound effect it has on the model’s internal reasoning dynamics.
> Unlike typical prefix tuning methods, often used for response-level control or refusal behaviors that degrade quickly after the first few tokens, SAFEPATH reshapes the model’s entire reasoning trajectory. No behavioral labels or reinforcement signals are applied. Yet, this minimal intervention leads to an emergent effect: the model internalizes the notion of safety and carries it throughout multi-step reasoning, rather than producing a shallow or one-time refusal.
> Most strikingly, the model spontaneously reactivates the Safety Primer mid-sequence when it encounters harmful or ambiguous contexts, indicating that SAFEPATH aligns the reasoning process, not just the output.
>
> > **Q2. Evaluation on recent attacks:** Suggestion to test SAFEPATH’s robustness against newer adversarial methods such as AutoDAN-Turbo and FlipAttack.
>
> We agree that evaluating SAFEPATH against more recent and sophisticated attack strategies is important for assessing its robustness. To address this, we have extended our evaluation to include FlipAttack [1], and ArtPrompt [4].
>
> Preliminary results show that SAFEPATH maintains strong performance across these newer attacks, flexibly recognizing and responding to harmful intent, rather than relying on rigid refusal templates, contributes to this robustness.
> We will include these new experimental results in the updated version of the paper to strengthen our empirical claims and further demonstrate SAFEPATH’s alignment effectiveness under diverse and evolving adversarial conditions.
>
> **Table: Additional Jailbreak Results (Attack Success Rate) on DeepSeek-R1-Distill-Llama-8B.**
> |Method|FLIP Attack ↓|ArtPrompt ↓|
> |-|-|-|
> |Base|45.2|26.0|
> |SAFEPATH|5.4|4.0|
>
> [1] Liu, Yue, et al. "Flipattack: Jailbreak llms via flipping." ICML (2025).\
> [2] Liu, Yue, et al. "Guardreasoner: Towards reasoning-based llm safeguards." ICLR FM-Wild Workshop. (2025).\
> [3] Wang, Cheng, et al. "Safety in large reasoning models: A survey." (2025).\
> [4] Jiang, Fengqing, et al. "Artprompt: Ascii art-based jailbreak attacks against aligned llms." ACL (2024).\
> [5] Ji, Jiaming, et al. "Beavertails: Towards improved safety alignment of llm via a human-preference dataset." NeurIPS (2023).\
> [6] Zou, Andy, et al. "Universal and transferable adversarial attacks on aligned language models." (2023).\
> [7] Jiang, Liwei, et al. "Wildteaming at scale: From in-the-wild jailbreaks to (adversarially) safer language models." NeurIPS (2024).

---

> > ### Comment · Reviewer_2uhd · 2025-08-02
> >
> > Thanks for your response.
> >
> > Most of my concerns have been solved. However, by checking other reviewers' comments, I agree with reviewer cJVT that `The defense baselines considered in this work are overly simplistic. The authors should compare against stronger and more relevant jailbreak defense methods, such as guard models like Llama-Guard, which offer plug-and-play capabilities and are more efficient than fine-tuning-based approaches.`
> >
> > It is reasonable since directly fine-tuning the base model may bring the safety tax. It is not clear which technical line is better in the practical deployment. Please at least compare your method with two kinds of guardrail models, like Llama Guard and GuardReasoner.
> >
> > I would like to raise my score if the authors can dispel the remaining concerns.

---

> > > ### Author Response · Authors · 2025-08-03
> > > **Response to Reviewer Concern**
> > >
> > > **Thank you sincerely for your thoughtful follow-up.** We understand and appreciate your concern regarding comparisons with guard models such as LLaMA Guard3 [1] and GuardReasoner [2].
> > >
> > > We acknowledge that guard models offer a practical and efficient solution for post-hoc safety filtering, particularly when direct access to the model is restricted. These methods operate externally, filtering or modifying inputs and outputs after generation. In contrast, SAFEPATH (SP) is an alignment-based approach designed to guide the internal reasoning trajectory of large reasoning models, focusing on in-context safety during multi-step generation.
> > >
> > > Because the two approaches operate at different levels of the safety pipeline, we believe they are complementary rather than directly comparable. That said, to fully address your concern, we conducted experiments comparing SAFEPATH to LLaMA Guard3 (LG) and GuardReasoner (GR), and additionally evaluated combinations of SAFEPATH with each. As shown below, SAFEPATH alone outperforms both guard models, and combining them further improves safety:
> > >
> > > **Table: SAFEPATH Compared to Additional Guard Model Baselines (8B)**
> > > |Method|BeaverTails↓|StrongReject↓|
> > > |-|-|-|
> > > |Base|37.3|35.0|
> > > |LLaMA Guard 3-8B (LG)|22.6|8.3|
> > > |GuardReasoner-8B (GR)|17.0|8.3|
> > > |**SAFEPATH (SP)**|**7.7**|**0.0**|
> > > |**LG+SP**|**7.5**|**0.0**|
> > > |**GR+SP**|**6.2**|**0.0**|
> > >
> > > We note that all methods, including SAFEPATH (as shown in Table 1 of our paper) and the guard model baselines, maintain reasoning accuracy similar to the base model, without any noticeable degradation.
> > > These results confirm that SAFEPATH achieves stronger safety performance than existing guard models, while targeting a complementary alignment objective. SAFEPATH can also be used alongside guard models to further enhance overall safety. We will include this discussion and corresponding results in the final version.
> > >
> > > [1] Dubey, Abhimanyu, et al. “The llama 3 herd of models.” (2024)\
> > > [2] Liu, Yue, et al. “Guardreasoner: Towards reasoning-based llm safeguards.” ICLR FM-Wild Workshop. (2025).

---

> > > > ### Comment · Reviewer_2uhd · 2025-08-03
> > > >
> > > > Thanks. Most of my concerns have been solved. It's interesting that the safety alignment method with the guard model can improve the safety of the whole LLM system.
> > > >
> > > > Minor suggestion:
> > > > - missing citation and discussion on recent hot paper [1]
> > > > - The original paper uses too many colors, especially in Figures 3, 4, 5, 6, and 8. The style could be simplified for better clarity.
> > > >
> > > > [1] Safety alignment should be made more than just a few tokens deep

---

> ### Author Response · Authors · 2025-08-03
> **Response to Reviewer Concern**
>
> **Thank you for the thoughtful comments.** We will cite and discuss [1] in the final version, as it highlights an important limitation of shallow, token-level alignment.
>
> SAFEPATH helps mitigate this issue by naturally embedding safety into the model’s internal reasoning process. Although the tuned prefix is short, the model frequently reactivates the Safety Primer mid-generation when encountering harmful content.
>
> In our analysis using partially completed harmful trajectories (following the setup in [1]), we observe that the Safety Primer is re-invoked over 15 times on average, showing that safety behavior is not tied to a fixed position but emerges dynamically as needed. This suggests that SAFEPATH enables the model to reason about safety throughout the generation process, rather than relying on static prompt cues.
>
> We also appreciate the suggestion regarding visual clarity and will simplify the color scheme in Figures 3, 4, 5, 6, and 8 to improve readability in the final version.
>
> [1] Qi, Xiangyu, et al. "Safety alignment should be made more than just a few tokens deep." ICLR (2025).

---

> > ### Author Response · Authors · 2025-08-09
> > **Gentle Reminder for Reviewer 2uhd**
> >
> > Dear Reviewer 2uhd,
> >
> > We sincerely appreciate the time and effort you dedicated to reviewing our paper. We've done our best to address your comments in the response and would be grateful for any additional thoughts before the discussion phase concludes. Thank you once again for your support.
> >
> > Sincerely, Authors

---

### Note · Authors · 2025-08-12

We sincerely thank reviewers for their thoughtful feedback and for recognizing SAFEPATH as a novel and efficient alignment method for large reasoning models (LRMs) that improves safety without sacrificing reasoning depth. Reviewers highlighted its ability to maintain a safe thinking path by reactivating safety guidance mid-reasoning when harmful content emerges, making it, to our knowledge, the first approach to directly harness an LRM's internal reasoning process.

A key strength is SAFEPATH's simplicity: it fine-tunes only the first 8 tokens while leaving the rest of the model untouched. Unlike prior "few-token" alignment methods that mainly affect the initial output, SAFEPATH shapes the entire reasoning trajectory, keeping safety guidance active through multi-step reasoning. This minimal intervention produces dynamic, context-aware safety without harming reasoning depth and can be applied at very low computational cost.

**2uhd** valued its complementarity with guard models, **N8gg** emphasized the safety–reasoning balance, **CSoy** noted the emergent dynamic safety, and **cJVT** acknowledged strengthened experimental evidence after rebuttal.

**Regarding concerns**

**Comparison with stronger baselines**

We added experiments against LLaMAGuard3, GuardReasoner, RealSafe-R1, and Star-R1, showing that SAFEPATH outperforms all of them.

**Evaluation scope**

We extended evaluations to FlipAttack and ArtPrompt, confirming robustness to more sophisticated attacks with consistently low attack success rates (ASRs).

**Method clarity and distinction from backdoor approaches**

We will revise Section 3 with a clearer pipeline diagram and explicitly note that SAFEPATH induces dynamic, context-aware safety reasoning rather than relying on memorized patterns.

**Generalization beyond DeepSeek-R1**

We evaluated SAFEPATH on s1.1, a non-distilled model, and found similar safety gains with preserved reasoning, showing the method is not tied to a specific architecture or training methods.

**Dynamic safety behavior and depth of alignment**

Under the MoreThink decoding strategy (adding wait to for reasoning expansion), SAFEPATH maintained reasoning performance while further lowering ASRs.

We believe our rebuttal provides concrete evidence addressing all concerns and further strengthens the case for SAFEPATH as a first-of-its-kind approach that achieves end-to-end safe reasoning with a minimal, 8-token intervention that shapes the model's entire reasoning process.

---

### Decision · Program_Chairs · 2025-09-17

**Decision:**

Accept (poster)

**Comment:**

This paper introduces SafePath, a lightweight alignment method that improves the safety of Large Reasoning Models (LRMs) when they encounter harmful prompts, while keeping the impact on reasoning performance small. All reviewers expressed consistent positive feedback on the proposed method, and the authors actively addressed many concerns during the rebuttal. This is a solid work, and I recommend acceptance.